# Risk factors associated with sexually transmitted infections and HIV among adolescents in a reference clinic in Madrid

**Oskar Ayerdi Aguirrebengoa**[1]*, **Mar Vera Garcia**[1], **Montserrat Rueda Sanchez**[2], **Giovanna D´Elia**[1], **Belén Chavero Méndez**[2], **María Alvargonzalez Arrancudiaga**[2], **Sandra Bello León**[2], **Teresa Puerta López**[1], **Petunia Clavo Escribano**[1], **Juan Ballesteros Martín**[1], **Blanca Menendez Prieto**[1], **Manuel Enrique Fuentes**[3], **Mónica García Lotero**[1], **Montserrat Raposo Utrilla**[1], **Carmen Rodríguez Martín**[1], **Jorge Del Romero Guerrero**[1]

1 Centro Sanitario Sandoval, Hospital Clínico San Carlos, IdISSC, Madrid, Spain, 2 Hospital Fundación Jiménez Díaz, Madrid, Spain, 3 Servicio Estadística, Hospital Clínico San Carlos, Madrid, Spain

* oskarayerdi@hotmail.com

## Abstract

### Introduction

Adolescents have a higher incidence of sexually transmitted infections (STIs) than persons of older age groups. The WHO emphasises the need to adopt specific and comprehensive prevention programmes aimed at this age group. The objective of this work was to analyse the prevalence of HIV/STIs among adolescents and to identify the sociodemographic, clinical and behavioural markers associated with these infections, in order to promote specific preventive strategies.

### Methodology

Retrospective descriptive study of adolescents, aged 10–19 years, who were attended to for the first consultation between 2016 and 2018 in a reference STI clinic in Madrid. All adolescents were given a structured epidemiological questionnaire where information on socio-demographic, clinical and behavioural characteristics was collected. They were screened for human inmmunodeficiency virus (HIV) and other sexually transmitted infections (STIs). The processing and analysis of the data was done using the STATA 15.0 statistical package.

### Results

The frequency of HIV/STIs detected among all adolescents was: gonorrhoea 21.7%, chlamydia 17.1%, syphilis 4.8% and HIV 2.4%. After conducting a multivariate analysis, the independent and statistically significant variables related to the presence of an STI were having first sexual relations at a young age and having a history of STIs. Latin American origin was just below the level of statistical significance (p = 0.066).

**Data Availability Statement:** All relevant data are within the manuscript and its Supporting Information files.

**Funding:** There was no funding for this study.

**Competing interests:** The authors have declared that no competing interests exist.

## Discussion/Conclusions

Adolescents who begin sexual relations at an early age or those who have a history of HIV/STIs are at higher risk of acquiring STIs. Comprehensive prevention programmes aimed specifically at adolescents should be implemented, especially before the age of 13 years.

## Introduction

Adolescence is considered to be the transitional age between childhood and adulthood, from 10 to 19 years [1]. Adolescents are at higher risk of acquiring sexually transmitted infections (STIs) compared to adults and they should be considered a special population in terms of STIs [2]. Many are having sexual relations at increasingly young ages and using alcohol and drugs during sex [3]. They may also face barriers of access to the healthcare system due to lack of awareness and knowledge, incompatibility of their schedule or concerns for their anonymity and confidentiality [4]. This carries an elevated risk of contracting an STI, including the human immunodeficiency virus (HIV). In addition, there are no guidelines relating to the frequency of STI screenings in adolescents.

Worldwide, it is estimated that 357.4 million cases of the four most common curable STIs occur annually: chlamydia (130.9 million cases), gonorrhoea (78.3 million), syphilis (5.6 million) and trichomoniasis (142.6 million) [5,6]. The annual epidemiological report by the European Center for Disease Control (ECDC) reports that, in the year 2016, 403,807 cases of chlamydia infection were reported in Europe, primarily among women aged 15–25 years, 2,043 cases of lymphogranuloma venereum (LGV), especially in men who have sex with men (MSM) older than 25 years, 75,349 episodes of gonorrhoea, which mainly affected MSM aged 20–34 years and women aged 15–19 years, and 29,365 cases of syphilis mostly in MSM older than 25 years[7]. In Spain, the reported STI incidence during 2017 for chlamydia was 24,55 cases per 100.000 people/year, gonorrhoea 18,74 cases per 100.000 people/year and syphilis 10,61 cases per 100.000 people/year, comparable to those described in Europe [8].

In 2017, there were 1.8 million adolescents living with HIV worldwide, which represents 5% of the total prevalence. Of those adolescents, 85% resided in sub-Saharan Africa [9]. In the same year, 250,000 new infections were diagnosed among persons aged 15–19 years, 16% of the total of new cases, the majority being in women of sub-Saharan Africa. In developed countries, new cases among adolescents are mostly diagnosed in men [10]. There were 25,353 new cases in the European Union, the majority in men, 38.2% were in MSM and 11.1% were between 15–24 years old [11]. In Spain, 3,381 new diagnoses of HIV were reported in 2017, 84.6% in men and the majority aged 25–34 years [12].

The World Health Organisation (WHO) highlights the need to adopt a comprehensive package of essential preventive interventions against HIV and other STIs [13]. Therefore, it is necessary to know updated epidemiological data in order to develop specific preventive strategies regarding sexual health[14].

The aim of this study was to analyse the prevalence of STIs/HIV among adolescents and to identify the sociodemographic, clinical and behavioural markers associated with these pathologies, in order to establish specific preventive measures.

## Methodology

### Study design and analysed population

Retrospective descriptive study in adolescents, aged 10–19 years, who were attended to for the first time between 1st of January of 2016 and 31th of December 2018 in a free and easily

accessible STI clinic located in Madrid. During this period, a total of 12,474 persons, with issues related to HIV / STIs, were seen to for the first consultation, aged 1–84 years, of which 3% (374) were adolescents.

## Variables

The data were obtained through a structured epidemiological questionnaire where information on sociodemographic, clinical and behavioural characteristics was collected: sex (men or women), age, sexual behavior (heterosexual men, men who have sex with men, women), origin (Spain, Latino America, Europe, Africa, Asia, North America), number of sexual partners year (0–5, 6–50, >50), number of sexual partners lifetime (1–10, 11–25, 26–100, >100), age of first sexual relations (≤13, 14–16, 17–19), type of sexual practices (oral sex, vaginal sex, insertive anal intercourse, receptive anal intercourse), frequency in the systematic use of condom (0%, <50%, ≥50%, 100%) or other preventive measures (post-exposure prophylaxis, pre-exposure prophylaxis), history of STIs, diagnoses of STIs at the time of first consultation (gonorrhea, chlamydia, syphilis, HIV), toxic habits (use of each drug, unprotected sexual practice that occurred under each effect: alcohol, tabacco, cannabis, cocaine, poppers, MDMA/ecstasy, ketamine, metanfetamine/crystal/tina, GHB, mephedrone), use of mobile applications in the search for sexual contacts and others (sex workers, victims of sexual abuse).

The following diagnostic tests were carried out based on the risk of acquiring HIV/STIs. HIV serologies (CMIA and Western Blot confirmation), syphilis (dark field microscopy, RPR, EIA and TPPA) and hepatotropic viruses: hepatitis A virus (HAV), hepatitis B virus (HBV) and hepatitis C virus (HCV) through chemiluminescent microparticle immunoassay (CMIA), Architect (Abbott). Past HBV infection is characterized by the presence of HBcAc and HBsAc with absence of HBsAg. Acute HBV infection is characterized by the presence of HBsAg and immunoglobulin M (IgM). During the initial phase of infection, patients are also seropositive for hepatitis B e antigen (HBeAg). Chronic infection is characterized by the persistence of HBsAg for at least 6 months (with or without concurrent HBeAg). Genital and extra-genital exudates were taken for the detection of: *Neisseria gonorrhoeae* (NG) by Gram staining, culture in Thayer Martin medium, NH API and PCR, and *Chlamydia trachomatis* (CT), PCR, and genotyped for lymphogranuloma venereum (LGV).

## Statistical analysis

The qualitative variables are shown with their frequency distribution. The quantitative variables are summarised with the average and standard deviation or with the median and interquartile range (IQR) if they do not fit within a normal distribution. The association between qualitative variables and the presence of an STI was carried out using the chi-squared test or Fisher's exact test, if necessary. A logistic regression model was adjusted with the objective of identifying the factors that are independently associated with the presence of an STI. The factors that were introduced in the logistic regression model were those that presented a p <0.10 in the bivariate analyze and/or clinically relevant. A significance level of 5% was accepted for all variables. The processing and analysis of the data was done using the STATA 15.0 statistical package.

## Ethic statement

The data were obtained through a structured epidemiological questionnaire systematically filled during the usual clinical practice. For the study, all data of the medical history were obtain fully anonymized before accessed them and the ethics committee waived the requirement for informed consent. The protocol was approved by the CEIC Hospital Clínico San Carlos, approval Number: 19/469 (S2).

## Results

Between January of 2016 and December of 2018, a total of 374 adolescents came to a reference STI clinic in Madrid for the first consultation. In 2016, 119 (31.8%) were attended to; in 2017, 111 (29.7%) and in 2018, 144 (38.5%).

Of the total number, 62.6% (234) were men. According to sexual behavior, 39.8% (149) were MSM, 22.7% (85) heterosexual men (HTX) and 37.4% (140) women (W) who had sex with men. Of these women, 5% (7) also had sex with women. The average age, of the three categories of transmission, was 17.9 years (± 1.1) with a minimum age of 13 years (Table 1).

Among the adolescents analysed, 42.8% (IQ95%:37.7–48.0) were diagnosed with an STI at the time of the first consultation, a total of 160. There were 110 adolescents with one STI, 41 with two and nine with three concomitant infections. Table 1 illustrates the frequencies of STIs according to sexual behavior. In adolescents diagnosed with STIs, 50.6% (81) had a gonorrhoea infection. There were 52 cases in MSM, of which 40.4% (21) were rectal, 38.5% (30) pharyngeal and 21.1% (11) urethral. In HTX, 100% were urethral. Among women, there were 21 cases, of which 66.7% (14) were cervical, 23.8% (5) pharyngeal and 9.5% (2) rectal. Of the 160, 40.0% (64) had chlamydia. Among MSM, there were 18 cases, 61.1% (11) were rectal, 27.8% (5) pharyngeal and 11.1% (2) urethral. All chlamydia in the HTX were urethral. In women, there were 28 cases, 85.7% (24) in the cervix and 14.3% (4) in the pharynx. There were no cases of LGV. In addition, 11.3% (18) presented syphilis: 21.1% (15) among MSM with STIs, 2.6% (1) of HTX and 4% (2) of women. Of those diagnosed with syphilis, 27.8% were detected in the primary phase, 38.9% secondary, 27.8% early latent and 5.6% late latent.

Regarding other STIs, 11.3% (18) had condyloma acuminatum, 7.5% (12) *Ureaplasma urealyticum* urethritis and 7.5% (12) anogenital herpes: 75% (9) genital and 25% (3) perianal. There were two cases of vaginal trichomoniasis (1.3%).

The prevalence of HIV among adolescents was 2.4% [9/372(IQR95%:1.1–4.15)], all MSM aged 18–19 years, with the exception of one who was 15. The prevalence of HIV among MSM was 7.4% (9/149). Two MSM who came to the first consultation with a previous diagnosis of HIV were not included, both in antiretroviral therapy (ART). The temporal development of the prevalence of HIV, among MSM, was: 8% (4/50) in 2016, 8.7% (4/46) in 2017 and 1.9% (1/53) in 2018. In 77.8%, information on the CD4 lymphocyte count was available upon diagnosis. The median CD4 count was 659 cells/ml (IQR: 626.5–669). There were no late diagnoses (<350 cells/ml CD4). Three of the nine diagnosed had no previous serologies. Of all MSM, two received postexposure prophylaxis (PEP) and one pre-exposure prophylaxis (PrEP) throughout their lives.

Regarding hepatitis B, 1.3% (4) had serological markers of a past hepatitis infection, 0.3% (1) acute and 0.3% (1) chronic; 16.7% (44) past hepatitis A and one hepatitis C cured. According to vaccination, 90.4% (281) vaccinated for HBV and 3.0% (8) for HAV.

Of all adolescents with STIs, 28.1% (45) were asymptomatic. Of genital gonorrhoea and chlamydia infections, 23.1% were asymptomatic as well as 55.7% of the extra-genital cases.

Table 2 analyses the frequency of drug use and unprotected sexual practices (USPs) that occurred under its effect. The 41.7% of MSM use psychotropic drugs, the 46.7% of HTX and 21.4% of women. Alcohol was the most frequent substance under which most USPs occurred. However, other substances such as methamphetamine, mephedrone or poppers, were associated with less condom use.

Table 3 shows the factors studied with the presence of STIs during the first consultation in the clinic among all adolescents. A multivariate analysis was conducted with the variables that presented a p<0.10 in the bivariate analysis (sexual behavior, origin, age of first sexual, number of sexual partners in the previous year, history of STIs and USPs under the effect of drugs).

**Table 1. Description of the sociodemographic, clinical and behavioural characteristics of adolescents at attended in the first consultation 2016–2018, according category of exposure (n = 374).**

| %(n) | | MSM 39.8 (149) | HTX 22.7 (85) | Women 37.4 (140) | Global 100 (374) |
|---|---|---|---|---|---|
| Origin | | | | | |
| | Spain | 72.5 (108) | 50,6 (43) | 53.6 (75) | 60.4 (226) |
| | Latino America | 18.1 (27) | 36.5 (31) | 31.4 (44) | 27.3 (102) |
| | Europe | 7.3 (11) | 4.8 (4) | 10 (14) | 7.8 (29) |
| | Africa | 2.0 (3) | 7.1 (6) | 2.9 (4) | 3.5 (13) |
| | Asia | 0 (0) | 1,2 (1) | 1,4 (2) | 0.8 (3) |
| | North America | 0 (0) | 0 (0) | 0.7 (1) | 0.3 (3) |
| Average age | | 18.1 (±0.9) | 18.0 (±1.1) | 17.7 (±1.2) | 17.9 (+/-1.1) |
| Age of first sexual relations | | | | | |
| | ≤13 | 12.75 (19) | 21.18 (18) | 8.57 (12) | 13.10 (49) |
| | 14–16 | 53.02 (79) | 61.18 (52) | 68.57 (96) | 60.70 (227) |
| | 17–19 | 30.87 (46) | 14.12 (12) | 17.86 (25) | 22.19 (83) |
| | Unknown | 3.36 (5) | 3.53 (3) | 5.00 (7) | 4.01 (15) |
| Number of sexual partners previous year | | | | | |
| | 0–5 | 49.66 (74) | 74.12(63) | 85.00 (119) | 68.45 (256) |
| | 6–50 | 32.89 (49) | 1.18 (1) | 0.00 (0) | 2.41 (9) |
| | >50 | 5.37 (8) | 14.12 (12) | 17.86 (25) | 2.41 (9) |
| | Unknown | 12.08 (18) | 7.06 (6) | 3.57 (5) | 7.75 (29) |
| Number of sexual partners/lifetime | | | | | |
| | 1–10 | 49.4 (74) | 56.4 (48) | 77.9 (109) | 61.7 (231) |
| | 11–25 | 16.1 (24) | 20.0 (17) | 11.4 (16) | 15.2 (57) |
| | 26–100 | 16.1 (24) | 10.6 (9) | 2.1 (3) | 9.6 (36) |
| | >100 | 6.0 (9) | 2.4 (2) | 0 (0) | 2.9 (11) |
| | Unknown | 12,1 (18) | 10,6 (9) | 8,6 (12) | 10,4 (39) |
| Sex workers | | | | | |
| | Yes | 6.04 (9) | 1.18 (1) | 3.57 (5) | 4.01 (15) |
| | No | 85.23 (127) | 82.35 (70) | 82.86(116) | 83.69 (313) |
| | Unknown | 8.72 (13) | 16.47 (14) | 13.57 (19) | 12.30 (46) |
| Victims of sexual abuse | | | | | |
| | Yes | 0.7 (1) | 0 (0) | 7.9 (11) | 3.2 (12) |
| | No | 66.4 (99) | 70,6 (60) | 62.9 (88) | 66.0 (247) |
| | Unknown | 32.9 (49) | 29.4 (25) | 29.3 (41) | 30.8(115) |
| History of STIs | | | | | |
| | Yes | 17.5 (29) | 11.8 (10) | 5.0 (7) | 12.3 (46) |
| | No | 80.54 (120) | 88.24 (75) | 95.00(133) | 87.70 (328) |
| Use of apps to find sexual relations | | | | | |
| | Yes | 53.69 (80) | 2.35 (2) | 2.14 (3) | 22.73 (85) |
| | No | 14.77(22) | 38.82 (33) | 39.29 (55) | 29.41 (110) |
| | Unknown | 31.54 (47) | 58.82 (50) | 58.57 (82) | 47.86 (179) |
| Diagnosed STIs | | 47.7 (71) | 45.9 (39) | 35.7 (50) | 42.8 (160) |
| | Gonorrhoea | 30.2 (45) | 22.4(19) | 12.2(17) | 21.7 (81) |
| | Chlamydia | 10.1 (15) | 25.9 (22) | 19.3 (27) | 17.1 (64) |
| | Syphillis | 10.1(15) | 1.2 (1) | 1.4 (2) | 4.8 (18) |
| | HIV | 7.4(9) | 0 | 0 | 2.4 (9) |

Fig 1 (Fig 1) illustrates the type of sexual practice and the systematic use of condom according to sexual behavior. As can be seen, there is little use of condom in all sexual practices, particularly notable in anal intercourse among heterosexuals and even lower in oral sex.

**Table 2. Analysis of the frequency of drug use and unprotected sexual practices (USPs) that occurred under its effect.**

| % (n) | Use %(n/315) | USPs under its effect %(n/Use) |
|---|---|---|
| Alcohol, tabacco and other drugs | 62.2 (196) | 27.55 (54) |
| • Alcohol | 42.5 (134) | 35.07 (47) |
| • Tabacco | 34.0 (107) | |
| • Psychotropic drugs | 34.9 (110) | 22.8 (23) |
| Cannabis | 25.71 (81) | 17.28 (14) |
| Cocaine | 3.81 (12) | 58.33 (7) |
| Poppers | 3.17 (10) | 90.00 (9) |
| MDMA/Ecstasy | 3.17 (10) | 50.00 (5) |
| Ketamine | 0.95 (3) | 66.67 (2) |
| Metanfetamine/Crystal/Tina | 0.63 (2) | 100.00 (2) |
| GHB | 0.63 (2) | 50.00 (1) |
| Mephedrone | 0.32 (1) | 100.0 (1) |

The variable, use of apps to find sexual relations, had not been included taking into account given the high number of unknown data. The independent and statistically significant variables related to the presence of an STI were: having first sexual relations at a young age and a history of STIs. Latin American origin was just below the level of statistical significance (p = 0.066).

## Discussion

In this study a high frequency of STIs has been observed in all sexual categories of transmission, mainly gonorrhoea and chlamydia. In the US, more than half of STIs occur among persons aged 15–24, despite the fact that they only represent 25% of the sexually active population [13]. The Centers for Disease Control and Prevention (CDC) reports chlamydia as the most prevalent bacterial STI in the US, its highest rate being in young women, also reflected in the female adolescents in our study. In our study, gonorrhoea and syphilis were more frequent among MSM just as in other studies [8]. The incidence of LGV in adolescents is low in other European regions, however in this study there was no cases [2].

The prevalence of HIV among the adolescents studied is lower than that found in some regions of the world. In sub-Saharan Africa it is 16%, with two out of three of cases being in women [15]. However, the figures resemble those found in developed countries and match with those of the most frequent sexual behavior [11]. All HIV-positive cases were MSM. The decline in new diagnoses between 2017 and 2018 is notable. These preliminary data suggest the decreasing trend in the incidence of HIV, as is the case in developed countries where combination HIV prevention programmes have been implemented [16]. In Spain, younger MSM could gain a greater preventive benefit by implementing new additional strategies such as pre-exposure prophylaxis (PrEP) [17].

In this study the use of condom is low in all sexual practices, especially in oral sex. Some individuals, particularly youth, may engage in oral sex instead of vaginal sex because they believe it to be less risky for STIs transmission. The risk for oral sex is lower compared with vaginal or anal sex, however unprotected oral sex is also associated with the acquisition and transmission of STDs infection [18].

Despite adolescents being a target population, there are no specific recommendations regarding the frequency of STI/HIV screening [19]. The high frequency of asymptomatic STIs makes screening for STIs solely based on clinical symptoms relatively ineffective [20]. As well

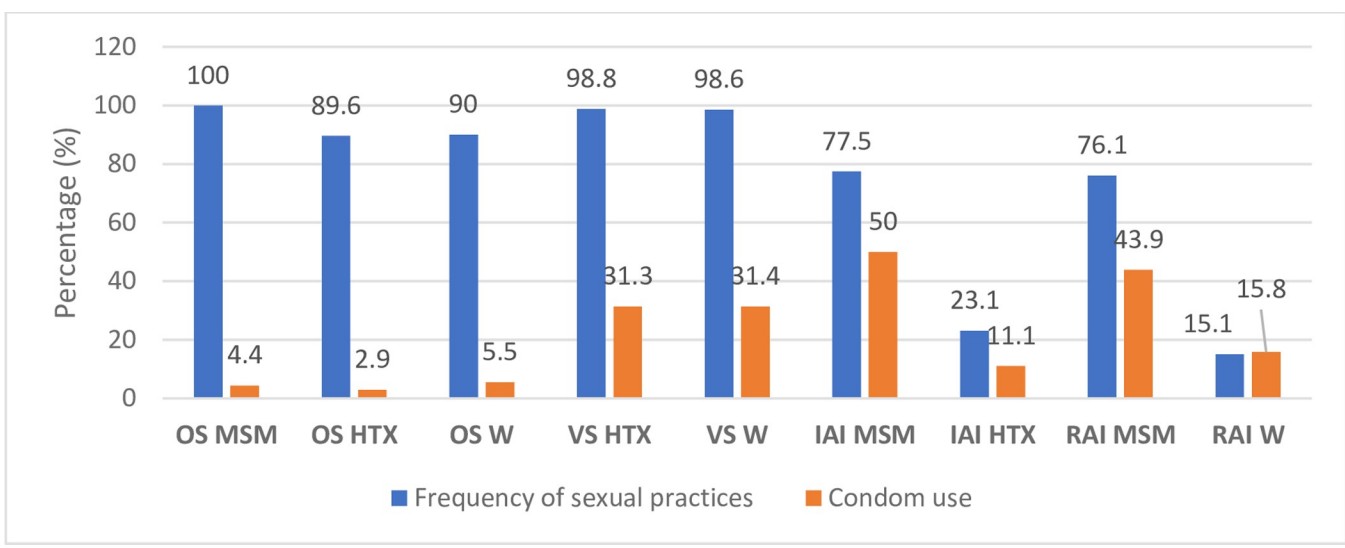

**Fig 1. Type of sexual practices and use of condom according to sexual behavior (N = 374).** Abbreviations: OS: oral sex; VS: vaginal sex; IAI: insertive anal intercourse and RAI: receptive anal intercourse.

as this, adolescents may be reluctant to report their sexual practices or may not consider their symptoms important through lack of awareness [21,22]. For these reasons, an STI/HIV screening should be considered, not only on clinical suspicion, but also through the identification of sociodemographic and behavioural markers.

The use of alcohol and drugs with sex reduces the perception of risk and tends toward unprotected sexual practices [23]. Yet, in our study, the use of these substances and the number of sexual partners in the previous year have not shown independent or statistically significant association with the presence of STIs. All cases of HIV were detected among MSM. However, the sexual behavior has not been associated as a risk indicator for the presence of other STIs, as is the case among adults [16]. By contrast, Latin American origin was close to statistical significance as an independent factor associated with the presence of STIs, possibly relating to poor healthcare resources and insufficient sex education [24].

Approximately 11% of the world's adolescents start having sexual relations before the age of 15 [25]. In our study, 9%-22% had their first sexual relations before the age of 14, according to sexual behavior. Beginning sexual relations at a young age, especially at 13 or younger, and the presence of a history of STIs are factors which have been independently associated with the presence of HIV/STIs, as has also been reported in other publications [26]. These indicators, found in our study, can help establish specific prevention programs in this age group. To optimise this preventive effort, it should be aimed at persons in the early phase of adolescence, between 10 to 13 years, which is considered a critical period for primary intervention, as from this age onwards the preventive messages offered have shown less impact [27,28].

In our study, the use of mobile applications in the search for sexual contacts has not been a factor associated with the presence of STIs [29]. However, the use of social networks for the screening and control of STIs has proven to be a cost-effective measure among adults in regions with a high prevalence of STIs. In addition to school or family education, technological advances may be ideal additional resources that incorporate preventive measures such as contact tracing or post-treatment control with a truly beneficial impact. [30,31]

The limitations of this work are that it is a descriptive and retrospective study carried out in a single STI/HIV center in Spain and that, despite the heterogeneity of the adolescents analysed,

**Table 3. Analysis of the factors associated with the presence of STIs among adolescents during the first consultation (n = 374).**

| | STI YES (n = 160) %(n) | STI NO (n = 214) %(n) | p Bivariate | OR (IC95%) Bivariate | OR (IC95%) Multivariate | p Multivariate |
|---|---|---|---|---|---|---|
| Sexual behavior | | | 0.099 | | | |
| MSM | 47.7 (71) | 52.3 (78) | | 1.64 (1.02–2.63) | 1.39 (0.77–2.50) | 0.269 |
| HTX | 45.9 (39) | 54.1 (46) | | 1.53 (0.88–2.64) | 1.17 (0.63–2.16) | 0.605 |
| W | 35.7 (50) | 64.3 (90) | | 1 | 1 | |
| Origin | | | 0.067 | | | |
| Spain | 38.5 (87) | 61.5 (139) | | 1 | 1 | |
| Latin America | 53.9 (55) | 46.1 (47) | | 1.87 (1.16–3.00) | 1.68 (0.97–2.93) | 0.066 |
| Europe | 37.9 (11) | 62.1 (18) | | 0.98 (0.44–2.17) | 1.18 (0.47–2.98) | 0.723 |
| Others | 41.2 (7) | 58.8 (10) | | 1.12 (0.41–3.05) | 1.13 (0.35–3.65) | 0.832 |
| Age of first sexual relations | | | <0.001 | | | |
| ≤13 | 69.4 (34) | 30.6 (15) | | 6.28 (2.88–13.69) | 5.39 (2.23–13.02) | 0.000 |
| 14–16 | 44.1 (100) | 55.9 (127) | | 2.18 (1.26–3.86) | 2.76 (1.47–5.18) | 0.002 |
| 17–19 | 26.5 (22) | 73.5 (22) | | 1 | 1 | |
| Number of sexual partners in the previous year | | | 0.011 | | | |
| 0–5 | 40.2 (103) | 59.8 (153) | | 1 | 1 | |
| 6–50 | 40 (32) | 60 (48) | | 0.99 (0.59–1.65) | 0.76 (0.42–1.37) | 0.361 |
| >50 | 88.9 (8) | 11.1 (1) | | 11.88 (1.47–96.44) | 5.85 (0.62–55.60) | 0.124 |
| History of STIs | | | 0.001 | | | |
| Yes | 65.2 (30) | 34.8 (16) | | 2.86 (1.50–5.45) | 1.39 (1.09–1.79) | 0.008 |
| No | 39.6 (130) | 60.4 (198) | | 1 | 1 | |
| USPs under the effect of drugs | | | 0.305 | | | |
| Yes | 52.2 (12) | 47.8 (11) | | 1.56 (0.66–3.66) | | |
| No | 41.2 (112) | 58.8 (160) | | 1 | | |
| Use of apps to find sexual relations | | | 0.031 | | | |
| Yes | 40.0 (34) | 60.0 (51) | | 1.95 (1.06–3.59) | | |
| No | 25.5 (28) | 74.5 (82) | | 1 | | |

it may not be possible to extrapolate its results to the general population. On the other hand, there are very few cohorts of adolescents in which studies of this type have been conducted.

Adolescents who have a history of STIs or who begin sexual relations at an early age are at higher risk of acquiring STI/HIV. Comprehensive prevention programmes aimed specifically at adolescents should be implemented, especially before the age of 13 years.

## Supporting information

**S1 File. Abstract in local language (Spanish).**
(PDF)

**S1 Database.**
(CSV)

## Author Contributions

**Conceptualization:** Oskar Ayerdi Aguirrebengoa, Mar Vera Garcia, Carmen Rodríguez Martín, Jorge Del Romero Guerrero.

**Data curation:** Oskar Ayerdi Aguirrebengoa, Mar Vera Garcia, Montserrat Rueda Sanchez, Giovanna D´Elia, Belén Chavero Méndez, María Alvargonzalez Arrancudiaga, Sandra Bello León, Teresa Puerta López, Petunia Clavo Escribano, Juan Ballesteros Martín, Blanca

Menendez Prieto, Manuel Enrique Fuentes, Mónica García Lotero, Montserrat Raposo Utrilla, Jorge Del Romero Guerrero.

**Formal analysis:** Oskar Ayerdi Aguirrebengoa, Montserrat Rueda Sanchez, Giovanna D´Elia, Belén Chavero Méndez, María Alvargonzalez Arrancudiaga, Sandra Bello León, Manuel Enrique Fuentes, Montserrat Raposo Utrilla.

**Investigation:** Oskar Ayerdi Aguirrebengoa, Manuel Enrique Fuentes.

**Methodology:** Oskar Ayerdi Aguirrebengoa, Manuel Enrique Fuentes, Montserrat Raposo Utrilla, Carmen Rodríguez Martín, Jorge Del Romero Guerrero.

**Project administration:** Oskar Ayerdi Aguirrebengoa, Mónica García Lotero, Montserrat Raposo Utrilla.

**Supervision:** Oskar Ayerdi Aguirrebengoa, Teresa Puerta López, Petunia Clavo Escribano, Juan Ballesteros Martín, Carmen Rodríguez Martín, Jorge Del Romero Guerrero.

**Validation:** Oskar Ayerdi Aguirrebengoa.

**Visualization:** Oskar Ayerdi Aguirrebengoa, Mar Vera Garcia.

**Writing – original draft:** Oskar Ayerdi Aguirrebengoa, Montserrat Rueda Sanchez, Giovanna D´Elia, Belén Chavero Méndez, María Alvargonzalez Arrancudiaga, Sandra Bello León, Juan Ballesteros Martín.

**Writing – review & editing:** Oskar Ayerdi Aguirrebengoa, Teresa Puerta López, Petunia Clavo Escribano, Manuel Enrique Fuentes, Carmen Rodríguez Martín, Jorge Del Romero Guerrero.

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
