## [Decision Letter · Decision Letter 0]

7 Jan 2020

PONE-D-19-33032

ADOLESCENTS, STIs AND HIV IN MADRID

PLOS ONE

Dear Dr. AYERDI,

Thank you for submitting your manuscript to PLOS ONE. After careful consideration, we feel that it has merit but does not fully meet PLOS ONE’s publication criteria as it currently stands. Therefore, we invite you to submit a revised version of the manuscript that addresses the points raised during the review process.

I agree with the comments made by the reviewers. Please pay specific attention to the following issues: the manuscript title should be revised to be more informative about the study content and the ethics approval and details of recruitment/consent procedures should be mentioned in the manuscript.

We would appreciate receiving your revised manuscript by Feb 21 2020 11:59PM. To enhance the reproducibility of your results, we recommend that if applicable you deposit your laboratory protocols in protocols.io, where a protocol can be assigned its own identifier (DOI) such that it can be cited independently in the future. For instructions see: http://journals.plos.org/plosone/s/submission-guidelines#loc-laboratory-protocols

We look forward to receiving your revised manuscript.

Kind regards,

Remco PH Peters, MD, PhD, DLSHTM

Academic Editor

PLOS ONE

Journal Requirements:

2. Please include additional information regarding the survey or questionnaire used in the study and ensure that you have provided sufficient details that others could replicate the analyses. If you developed and/or translated a questionnaire as part of this study and it is not under a copyright more restrictive than CC-BY, please include a copy, in both the original language and English, as Supporting Information.

3. In the ethics statement in the manuscript and in the online submission form, please provide additional information about the patient records/samples used in your retrospective study.

Specifically, please ensure that you have discussed whether all data/samples were fully anonymized before you accessed them and/or whether the IRB or ethics committee waived the requirement for informed consent.

If patients' parents/guardians provided informed written consent to have data/samples from their medical records used in research, please include this information

3. Thank you for including your ethics statement: 

"Ethic Committee approval number: 19/469-E"

Please provide an amended Funding Statement that declares *all* the funding or sources of support received during this specific study (whether external or internal to your organization) as detailed online in our guide for authors at http://journals.plos.org/plosone/s/submit-nowPlease state what role the funders took in the study.  If any authors received a salary from any of your funders, please state which authors and which funder. If the funders had no role, please state: "The funders had no role in study design, data collection and analysis, decision to publish, or preparation of the manuscript."

6. Please include captions for your Supporting Information files at the end of your manuscript, and update any in-text citations to match accordingly. Please see our Supporting Information guidelines for more information: http://journals.plos.org/plosone/s/supporting-information

Reviewers' comments:

Reviewer's Responses to Questions

**Comments to the Author**

1. Is the manuscript technically sound, and do the data support the conclusions?

Reviewer #1: Yes

Reviewer #2: Yes

2. Has the statistical analysis been performed appropriately and rigorously? 

Reviewer #1: Yes

Reviewer #2: Yes

3. Have the authors made all data underlying the findings in their manuscript fully available?

Reviewer #1: No

Reviewer #2: Yes

4. Is the manuscript presented in an intelligible fashion and written in standard English?

Reviewer #1: No

Reviewer #2: No

5. Review Comments to the Author

Reviewer #1: General comments

The study presented by Ayerdi Oskar et al., is an interesting study whose objective has been to analyze the prevalence of STI / HIV among adolescents and identify the sociodemographic, clinical and behavioral markers associated with these pathologies, in order to establish specific preventive measures . It is a novel study that provides information on this highly vulnerable population group and also assesses the importance of early detection of these STIs not only based on symptomatic parameters but also on sociodemographic and behavioral variables, as they are many of these asymptomatic infections. . In general, the format of the tables is very improved, being necessary to correct arithmetic errors, and to unify criteria when presenting the results of the same, the limitation of the study regarding being a descriptive study with the participation of a single center and the difficulty of extrapolating the results to the entire population is perfectly indicated in the text.

Specific comments

1. Title: The title is very short and not very informative, not reflecting the objective of the article properly, on the other hand, precisely because it is very short it makes no sense to put acronyms on it and not put the words with its full name, it is not correct to put acronyms on the titles of scientific works, unless strictly necessary, this being not the case.

2. Authors: Powerfully draws attention to the high number of authors for an article whose methodology is simple, it would be convenient to justify the reason for this decision.

3. Summary: In the methodology section the acronym "HIV / STIs: HIV, HAV, HBV, HCV", being the first time they are cited in the text must go with the full word and in brackets its corresponding acronym.

4. Introduction: As in the previous case, for the acronym “ECDC” the first time the complete word with its acronyms should be named in parentheses.

5. Methodology:

- The ethical aspects are not described, was the study reviewed and approved by an Ethics Committee?

- In the regression they propose, they do not explain what procedures they followed to analyze the regression adjustment criteria performed.

- When “structured epidemiological questionnaire” is mentioned in the variables, it is not mentioned if said questionnaire was prepared exclusively for the sample of adolescent population or if it is the questionnaire that is systematically filled out for the entire population that attends the STI center of Madrid .

- When talking about “transmission category” it would be more correct to talk about “sexual behavior”. Apply this throughout the entire text.

6. Results:

- Table 1. “Description of the sociodemographic, clinical and behavioral characteristics of adolescents at atended in the first consultation 2016-2018, according to category of exposure”:

• Review the numerical results of the “Global” column as errors are detected.

• Regarding the variables “Post-exposure prophylaxis” (PEP) and “Pre-exposure prophylaxis” (PrEP) the numerical figures that appear in the MSM exposure category are reversed, instead of% (n) they are n (% ).

• “Oral contraceptives”, I do not think it is necessary to include this in the table since the use of oral contraceptives protects from an unwanted pregnancy but not from the acquisition and / or transmission of its which is what the article is about.

Figure 1. “Type of sexual practices and use of condom according to transmission category”:

• "As can be seen, there is little use of condom in all sexual practices, particularly notable in anal intercourse among heterosexuals", it should also be added that it is even lower under condom use in the case of oral sex (5.5 %, 2.9% and 4.4%).

• It is striking that there is VS in MSM (vaginal sex in men who have sex with men), it would be interesting to clarify whether in the category MSM men are also considered, although sometimes they have sex with men other times they have it with women.

- Table 2. “Analysis of the frequency of drug use and unprotected sexual practices (USPs) that occurred under its effect”:

• It is necessary to improve the format of the table, it is not understood why you put "psycotropics drugs" and then again when you talk about type of drugs you put "psycotropics drugs" again, in both cases the assigned numerical values do not match.

• You have to unify criteria in the tables: in table 1 the legend is at the foot of the table and in table 2 the legend is in the headboard and in table 3 the legend is again at the foot of the table.

- Table 3. “Analysis of the factors associated with the presence of STIs among adolescents during the first consultation (n = 374)”:

• “A multivariate analysis was conducted with the variables that presented a p <0.10 in the univariate analysis ……, it is a mistake and it is not univariate but bivariate, it is the bivariate analysis that leads to a multivariate analysis to confirm the statistical significance of the variables eliminating confusion factors.

• Where are the results of that bivariate analysis?

• Again, criteria must be unified: the p values of the multivariate analysis when they do not give significance or are all put in a middle dash or left blank.

7. Discussion:

- Acronym “CDC”: idem as explained above: “Centers for Disease Control and Prevention” (CDC).

-”Likewise, the low incidence of LGV in adolescents matches with other European regions”, does not coincide with the results section when it says: “there were no cases of LGV”, so there was no low incidence directly there was not.

- Regarding the scarce use of condoms in oral sex, it is necessary to comment in this section that oral sex is also a form of its transmission, and this statement must be referenced.

8. Bibliography: The regulations of the journal regarding how to write bibliographic references in Vancouver style must be thoroughly reviewed and criteria must be unified, in general: in the online documents there are many links to the internet and the date of access, as with the "doi", capital letters when they are not necessary, links that are wrong or that give you an error "page not found".

Reviewer #2: This is a very interesting study that provides data on STIs in adolescents and their potential risk behaviours, a topic on which there is little literature to date. However, there are some specific areas for improvement which I suggest to review.

Abstract - Methods: As noted in the manuscript, other diagnostic tests are carried out in addition to those mentioned. Since this is the abstract, I suggest not to specify any of them.

Introduction:

- Line number 4: add a reference.

- Second paragraph, first line: adding “curable” after “four most common” would be more accurate.

- It would be interesting to provide the data on the incidence of STIs in Spain (as in the previous lines total data are reported, not incidences).

Methods:

- Study design...: Specify exact dates. Describe the potential users of the clinic (age range).

- I suggest explaining the diagnostic tests in a separate section. I think that diagnostic tests require a broader explanation, (techniques and protocol followed to request them). The authors should mention all the diagnostic tests included. The technique used for CD4 should also be recorded. Definition of past/acute/chronic hepatitis B should be included.

Results:

- If data are available, it would be interesting to know the reason for consultation.

- Last line of second paragraph: The maximum age can be ignored since it is an inclusion criterion.

- Table 1: Some figures do not seem to be correct (the sum of the parts in some columns is greater than the total, for example). On the other hand, I think it would be useful to add a row with the number of patients for whom the data is unknown.

- I deduce that some of the patients have more than one STI, I suggest specifying it in the text.

- Table 2: “Psychotropic drugs” appears twice.

Discussion:

- Last sentence of first paragraph: The authors should not comment on the “low” incidence of LGV as it is zero.

- I suggest placing third paragraph at the end.

Figure1: MSM instead of HSH.

Finally, I recommend language and writing editing. The authors should review the use of abbreviations and the writing of the species names according to the journal guidelines. In the text, reference numbers should be cited in square brackets. Moreover, the authors should review the format of the references.

6. PLOS authors have the option to publish the peer review history of their article (what does this mean?). If published, this will include your full peer review and any attached files.

Reviewer #1: No

Reviewer #2: No

---

## [Author Response · Author response to Decision Letter 0]

24 Jan 2020

Dear Editor and Reviewers,

Thank you for considering our manuscript to PLOS ONE. After careful review, we have submited the 'Manuscript', 'Revised Manuscript with Track Changes', 'Response to Reviewers' and additional supporting information. 

Journal Requirements:

Updated.

2. Please include additional information regarding the survey or questionnaire used in the study and ensure that you have provided sufficient details that others could replicate the analyses. If you developed and/or translated a questionnaire as part of this study and it is not under a copyright more restrictive than CC-BY, please include a copy, in both the original language and English, as Supporting Information.

A special questionnaire was not used to the study. The data were obtained through a structured epidemiological questionnaire systematically filled during the usual clinical practice (structured medical history). Tha variables have been included in the manuscript. Information on sociodemographic, clinical and behavioural characteristics was collected: sex (men or women), age, sexual behavior (heterosexual men, men who have sex with men, women), origin (Spain, Latino America, Europe, Africa, Asia, North America), number of sexual partners year (0-5, 6-50, >50), number of sexual partners lifetime (1-10, 11-25, 26-100, >100), age of first sexual relations (≤13, 14-16, 17-19), type of sexual practices (oral sex, vaginal sex, insertive anal intercourse, receptive anal intercourse), frequency in the systematic use of condom (0%, <50%, ≥50%, 100%) or other preventive measures (post-exposure prophylaxis, pre-exposure prophylaxis), history of STIs, diagnoses of STIs at the time of first consultation (gonorrhea, chlamydia, syphilis, HIV), toxic habits (use of each drug, unprotected sexual practice that occurred under each effect: alcohol, tabacco, cannabis, cocaine, poppers, MDMA/ecstasy, ketamine, metanfetamine/crystal/tina, GHB, mephedrone), use of mobile applications in the search for sexual contacts and others (sex workers, victims of sexual abuse).

3. In the ethics statement in the manuscript and in the online submission form, please provide additional information about the patient records/samples used in your retrospective study.

Specifically, please ensure that you have discussed whether all data/samples were fully anonymized before you accessed them and/or whether the IRB or ethics committee waived the requirement for informed consent.

If patients' parents/guardians provided informed written consent to have data/samples from their medical records used in research, please include this information

The data were obtained through a structured epidemiological questionnaire systematically filled during the usual clinical practice. For the study, all data of the medical history were obtain fully anonymized before accessed them and the ethics committee waived the requirement for informed consent. The protocol was approved by the CEIC Hospital Clínico San Carlos, approval Number: 19/469 (S2).

3. Thank you for including your ethics statement: 

"Ethic Committee approval number: 19/469-E"

a) Please amend your current ethics statement to include the full name of the ethics committee/institutional review board(s) that approved your specific study. Included.

b) Once you have amended this/these statement(s) in the Methods section of the manuscript, please add the same text to the “Ethics Statement” field of the submission form (via “Edit Submission”). Included.

a. Please provide an amended Funding Statement that declares *all* the funding or sources of support received during this specific study (whether external or internal to your organization) as detailed online in our guide for authors at http://journals.plos.org/plosone/s/submit-now

b. Please state what role the funders took in the study. If any authors received a salary from any of your funders, please state which authors and which funder. If the funders had no role, please state: "The funders had no role in study design, data collection and analysis, decision to publish, or preparation of the manuscript." There was no funders for this study.

c. Please include your amended statements within your cover letter; we will change the online submission form on your behalf. Included.

5. Please include a separate caption for each figure in your manuscript. Included.

6. Please include captions for your Supporting Information files at the end of your manuscript, and update any in-text citations to match accordingly. Please see our Supporting Information guidelines for more information: http://journals.plos.org/plosone/s/supporting-information Included.

Reviewers' comments:

Reviewer's Responses to Questions

Comments to the Author

1. Is the manuscript technically sound, and do the data support the conclusions?

Reviewer #1: Yes

Reviewer #2: Yes

2. Has the statistical analysis been performed appropriately and rigorously? 

Reviewer #1: Yes

Reviewer #2: Yes

 3. Have the authors made all data underlying the findings in their manuscript fully available?

Reviewer #1: No

Reviewer #2: Yes

We will include the database in the supporting information section so that it is fully available.________________________________________

4. Is the manuscript presented in an intelligible fashion and written in standard English?

Reviewer #1: No

Reviewer #2: No

The manuscript has been reviewed by a native translator from the United Kingdom.________________________________________

5. Review Comments to the Author

Reviewer #1: General comments

The study presented by Ayerdi Oskar et al., is an interesting study whose objective has been to analyze the prevalence of STI / HIV among adolescents and identify the sociodemographic, clinical and behavioral markers associated with these pathologies, in order to establish specific preventive measures . It is a novel study that provides information on this highly vulnerable population group and also assesses the importance of early detection of these STIs not only based on symptomatic parameters but also on sociodemographic and behavioral variables, as they are many of these asymptomatic infections. . In general, the format of the tables is very improved, being necessary to correct arithmetic errors, and to unify criteria when presenting the results of the same, the limitation of the study regarding being a descriptive study with the participation of a single center and the difficulty of extrapolating the results to the entire population is perfectly indicated in the text.

Specific comments

1. Title: The title is very short and not very informative, not reflecting the objective of the article properly, on the other hand, precisely because it is very short it makes no sense to put acronyms on it and not put the words with its full name, it is not correct to put acronyms on the titles of scientific works, unless strictly necessary, this being not the case.

The new title proposed: Frequency of sexually transmitted infections/HIV and the associated risk factors among adolescents in a refence STI Clinic of Madrid 

2. Authors: Powerfully draws attention to the high number of authors for an article whose methodology is simple, it would be convenient to justify the reason for this decision.

The main reason could be that the study participants have been attended by several doctors of the center. The contribution of each co-author is described and also included as supporting information.

Contributorship statemet: 

Oskar Ayerdi Aguirrebengoa: Substantial contributions to the conception or design of the work; or the acquisition, analysis, or interpretation of data for the work. Drafting the work or revising it critically for important intellectual content; AND Final approval of the version to be published; AND Agreement to be accountable for all aspects of the work in ensuring that questions related to the accuracy or integrity of any part of the work are appropriately investigated and resolved.

Mar Vera García: Substantial contributions to the acquisition of data for the work. Revising it critically for important intellectual content. Final approval of the version to be published. Agreement to be accountable for all aspects of the work in ensuring that questions related to the accuracy or integrity of any part of the work are appropriately investigated and resolved.

Montserrat Rueda Sanchez: Substantial contributions to the acquisition of data for the work. Drafting the work and revising it critically for important intellectual content. Final approval of the version to be published. Agreement to be accountable for all aspects of the work in ensuring that questions related to the accuracy or integrity of any part of the work are appropriately investigated and resolved.

Giovanna D´Elia: Substantial contributions to the acquisition of data for the work. Drafting the work and revising it critically for important intellectual content. Final approval of the version to be published. Agreement to be accountable for all aspects of the work in ensuring that questions related to the accuracy or integrity of any part of the work are appropriately investigated and resolved.

Belén Chavero Méndez: Substantial contributions to the acquisition of data for the work. Drafting the work and revising it critically for important intellectual content. Final approval of the version to be published. Agreement to be accountable for all aspects of the work in ensuring that questions related to the accuracy or integrity of any part of the work are appropriately investigated and resolved.

María Alvargonzalez Arrancudiaga: Substantial contributions to the acquisition of data for the work. Drafting the work and revising it critically for important intellectual content. Final approval of the version to be published. Agreement to be accountable for all aspects of the work in ensuring that questions related to the accuracy or integrity of any part of the work are appropriately investigated and resolved.

Sandra Bello León: Substantial Substantial contributions to the acquisition of data for the work. Drafting the work and revising it critically for important intellectual content. Final approval of the version to be published. Agreement to be accountable for all aspects of the work in ensuring that questions related to the accuracy or integrity of any part of the work are appropriately investigated and resolved.

Teresa Puerta López: Substantial contributions to the acquisition of data for the work. Revising it critically for important intellectual content. Final approval of the version to be published. Agreement to be accountable for all aspects of the work in ensuring that questions related to the accuracy or integrity of any part of the work are appropriately investigated and resolved.

Petunia Clavo Escribano: Substantial contributions to the acquisition of data for the work. Revising it critically for important intellectual content. Final approval of the version to be published. Agreement to be accountable for all aspects of the work in ensuring that questions related to the accuracy or integrity of any part of the work are appropriately investigated and resolved.

Juan Ballesteros Martín: Substantial contributions to the acquisition of data for the work. Revising it critically for important intellectual content. Final approval of the version to be published. Agreement to be accountable for all aspects of the work in ensuring that questions related to the accuracy or integrity of any part of the work are appropriately investigated and resolved.

Blanca Menendez Prieto: Substantial contributions to the acquisition of data for the work. Revising it critically for important intellectual content. Final approval of the version to be published. Agreement to be accountable for all aspects of the work in ensuring that questions related to the accuracy or integrity of any part of the work are appropriately investigated and resolved.

Manuel Enrique Fuentes Ferrer: Substantial contributions to design of the work; analysis and interpretation of data for the work; revising it critically for important intellectual content. Final approval of the version to be published. Agreement to be accountable for all aspects of the work in ensuring that questions related to the accuracy or integrity of any part of the work are appropriately investigated and resolved.

Mónica García Lotero: Substantial contributions to the acquisition of data for the work. Revising it critically for important intellectual content. Final approval of the version to be published. Agreement to be accountable for all aspects of the work in ensuring that questions related to the accuracy or integrity of any part of the work are appropriately investigated and resolved.

Montserrat Raposo Utrilla: Substantial contributions to the conception or design of the work; and the analysis and interpretation of data for the work. Revising it critically for important intellectual content. Final approval of the version to be published. Agreement to be accountable for all aspects of the work in ensuring that questions related to the accuracy or integrity of any part of the work are appropriately investigated and resolved.

Carmen Rodríguez Martín: Substantial contributions to the conception or design of the work. Drafting the work or revising it critically for important intellectual content. Final approval of the version to be published. Agreement to be accountable for all aspects of the work in ensuring that questions related to the accuracy or integrity of any part of the work are appropriately investigated and resolved.

Jorge Del Romero Guerrero: Substantial contributions to the conception or design of the work and the acquisition of data for the work. Drafting the work or revising it critically for important intellectual content. Final approval of the version to be published. Agreement to be accountable for all aspects of the work in ensuring that questions related to the accuracy or integrity of any part of the work are appropriately investigated and resolved.

3. Summary: In the methodology section the acronym "HIV / STIs: HIV, HAV, HBV, HCV", being the first time they are cited in the text must go with the full word and in brackets its corresponding acronym. Updated.

4. Introduction: As in the previous case, for the acronym “ECDC” the first time the complete word with its acronyms should be named in parentheses. Updated.

5. Methodology:

- The ethical aspects are not described, was the study reviewed and approved by an Ethics Committee? All data were obtain fully anonymized before accessed them and the ethics committee waived the requirement for informed consent. The protocol was approved by the CEIC Hospital Clínico San Carlos, approval Number: 19/469 (S2).

- In the regression they propose, they do not explain what procedures they followed to analyze the regression adjustment criteria performed.

The factors that were introduced in the logistic regression model were those that presented a p <0.10 (which was explained in results) in the bivariate analyze and/or clinically relevant. 

This sentence is added in the paragraph of statistical analysis.

- When “structured epidemiological questionnaire” is mentioned in the variables, it is not mentioned if said questionnaire was prepared exclusively for the sample of adolescent population or if it is the questionnaire that is systematically filled out for the entire population that attends the STI center of Madrid.The information obtained through the questionnaire or complete sexual history is the one is used in the center to create the medical history methodically. A specific questionnaire has not been created for this study. In the methodology all the variables that have been taken into account have been included

- When talking about “transmission category” it would be more correct to talk about “sexual behavior”. Apply this throughout the entire text.Updated.

6. Results:

- Table 1. “Description of the sociodemographic, clinical and behavioral characteristics of adolescents at atended in the first consultation 2016-2018, according to category of exposure”:

• Review the numerical results of the “Global” column as errors are detected. The unknown value has been described and all the table 1. data has been updated. 

• Regarding the variables “Post-exposure prophylaxis” (PEP) and “Pre-exposure prophylaxis” (PrEP) the numerical figures that appear in the MSM exposure category are reversed, instead of% (n) they are n (% ). Regarding PEP and PrEP, it is decided to give these information in the text.

• “Oral contraceptives”, I do not think it is necessary to include this in the table since the use of oral contraceptives protects from an unwanted pregnancy but not from the acquisition and / or transmission of its which is what the article is about. Removed.

Figure 1. “Type of sexual practices and use of condom according to transmission category”:

• "As can be seen, there is little use of condom in all sexual practices, particularly notable in anal intercourse among heterosexuals", it should also be added that it is even lower under condom use in the case of oral sex (5.5 %, 2.9% and 4.4%). Updated.

• It is striking that there is VS in MSM (vaginal sex in men who have sex with men), it would be interesting to clarify whether in the category MSM men are also considered, although sometimes they have sex with men other times they have it with women. Updated.

- Table 2. “Analysis of the frequency of drug use and unprotected sexual practices (USPs) that occurred under its effect”:

• It is necessary to improve the format of the table, it is not understood why you put "psycotropics drugs" and then again when you talk about type of drugs you put "psycotropics drugs" again, in both cases the assigned numerical values do not match. Updated. 

To facilitate compression, part of the information has been passes to the text.

• You have to unify criteria in the tables: in table 1 the legend is at the foot of the table and in table 2 the legend is in the headboard and in table 3 the legend is again at the foot of the table. Unified.

- Table 3. “Analysis of the factors associated with the presence of STIs among adolescents during the first consultation (n = 374)”:

• “A multivariate analysis was conducted with the variables that presented a p <0.10 in the univariate analysis ……, it is a mistake and it is not univariate but bivariate, it is the bivariate analysis that leads to a multivariate analysis to confirm the statistical significance of the variables eliminating confusion factors. The multivariate analysis was conducted with the variables that presented a p<0.10 in the biivariate analysis. This variables have been included in the text: sexual behavior, origin, age of first sexual, number of sexual partners in the previous year, history of STIs and USPs under the effect of drugs.The variable, use of apps to find sexual relations, had not been included taking into account given the high number of unknown data.

• Where are the results of that bivariate analysis? In the 4th and 5th columns have been included “bivariate”. 

• Again, criteria must be unified: the p values of the multivariate analysis when they do not give significance or are all put in a middle dash or left blank. Unified.

7. Discussion:

- Acronym “CDC”: idem as explained above: “Centers for Disease Control and Prevention” (CDC). Updated.

-”Likewise, the low incidence of LGV in adolescents matches with other European regions”, does not coincide with the results section when it says: “there were no cases of LGV”, so there was no low incidence directly there was not. Updated.

- Regarding the scarce use of condoms in oral sex, it is necessary to comment in this section that oral sex is also a form of its transmission, and this statement must be referenced. Updated.

8. Bibliography: The regulations of the journal regarding how to write bibliographic references in Vancouver style must be thoroughly reviewed and criteria must be unified, in general: in the online documents there are many links to the internet and the date of access, as with the "doi", capital letters when they are not necessary, links that are wrong or that give you an error "page not found". All the bibliography has been revised and updated.

Reviewer #2: This is a very interesting study that provides data on STIs in adolescents and their potential risk behaviours, a topic on which there is little literature to date. However, there are some specific areas for improvement which I suggest to review.

Abstract - Methods: As noted in the manuscript, other diagnostic tests are carried out in addition to those mentioned. Since this is the abstract, I suggest not to specify any of them. Updated.

Introduction:

- Line number 4: add a reference. Updated.

- Second paragraph, first line: adding “curable” after “four most common” would be more accurate. Updated.

- It would be interesting to provide the data on the incidence of STIs in Spain (as in the previous lines total data are reported, not incidences). Updated.

Methods:

- Study design...: Specify exact dates. Describe the potential users of the clinic (age range). Updated.

- I suggest explaining the diagnostic tests in a separate section. I think that diagnostic tests require a broader explanation, (techniques and protocol followed to request them). The authors should mention all the diagnostic tests included. The technique used for CD4 should also be recorded. Definition of past/acute/chronic hepatitis B should be included.The determination of lymphocyte subpopulations T CD4 + / T CD8 + was performed by flow cytometry in Aquios CL (Beckman Coulter). We have not included this information in the methodology since this study has not analyzed the immunological situation of the patients HIV diagnosed. Rest of the suggestions have been updated. 

Results:

- If data are available, it would be interesting to know the reason for consultation. All of them went to the center for issues related to HIV / STIs, included in the manuscript. 

- Last line of second paragraph: The maximum age can be ignored since it is an inclusion criterion. Updated.

- Table 1: Some figures do not seem to be correct (the sum of the parts in some columns is greater than the total, for example). On the other hand, I think it would be useful to add a row with the number of patients for whom the data is unknown. The unknown value has been described and all the table 1. data has been updated.

- I deduce that some of the patients have more than one STI, I suggest specifying it in the text. Included in the text.

- Table 2: “Psychotropic drugs” appears twice. Updated.

Discussion:

- Last sentence of first paragraph: The authors should not comment on the “low” incidence of LGV as it is zero. Updated.

- I suggest placing third paragraph at the end. It is a very interesting suggestion and we have commented for a long time among the drafting team. If you agree, we would prefer to leave it in that order for the moment. However, we believe that it is a possibility so we are available to make the change.

Figure1: MSM instead of HSH. Updated.

Finally, I recommend language and writing editing. The authors should review the use of abbreviations and the writing of the species names according to the journal guidelines. In the text, reference numbers should be cited in square brackets. Moreover, the authors should review the format of the references. square brackets. Reviewed.

6. PLOS authors have the option to publish the peer review history of their article (what does this mean?). If published, this will include your full peer review and any attached files.

Do you want your identity to be public for this peer review? For information about this choice, including consent withdrawal, please see our Privacy Policy.

Reviewer #1: No

Reviewer #2: No

In compliance with data protection regulations, you may request that we remove your personal registration details at any time.(Remove my information/details). Please contact the publication office if you have any questions.

---

## [Editor Report · Decision Letter 1]

29 Jan 2020

RISK FACTORS ASSOCIATED WITH SEXUALLY TRANSMITTED INFECTIONS AND HIV AMONG ADOLESCENTS IN A REFERENCE CLINIC IN MADRID

PONE-D-19-33032R1

Dear Dr. AYERDI,

We are pleased to inform you that your manuscript has been judged scientifically suitable for publication and will be formally accepted for publication once it complies with all outstanding technical requirements.

With kind regards,

Remco PH Peters, MD, PhD, DLSHTM

Academic Editor

PLOS ONE
---

## [Editor Report · Acceptance letter]

24 Feb 2020

PONE-D-19-33032R1 

Risk factors associated with sexually transmitted infections and HIV among adolescents in a reference clinic in Madrid  

Dear Dr. Ayerdi Aguirrebengoa:

I am pleased to inform you that your manuscript has been deemed suitable for publication in PLOS ONE. Congratulations! Your manuscript is now with our production department. 

With kind regards,

on behalf of

Prof Remco PH Peters 

Academic Editor

PLOS ONE